**∂ | Open Peer Review** | Bacteriology | Research Article

# Bacterial multispecies interaction mechanisms dictate biogeographic arrangement between the oral commensals *Corynebacterium matruchotii* and *Streptococcus mitis*

Eric Almeida,[1] Surendra Puri,[2] Alex Labossiere,[1] Subashini Elangovan,[2] Jiyeon Kim,[2] Matthew Ramsey[1]

**ABSTRACT**    Polymicrobial biofilms are present in many environments particularly in the human oral cavity where they can prevent or facilitate the onset of disease. While recent advances have provided a clear picture of both the constituents and their biogeographic arrangement, it is still unclear what mechanisms of interaction occur between individual species in close proximity within these communities. In this study, we investigated two mechanisms of interaction between the highly abundant supragingival plaque (SUPP) commensal *Corynebacterium matruchotii* and *Streptococcus mitis* which are directly adjacent/attached *in vivo*. We discovered that *C. matruchotii* enhanced the fitness of streptococci dependent on its ability to detoxify streptococcal-produced hydrogen peroxide and its ability to oxidize lactate also produced by streptococci. We demonstrate that the fitness of adjacent streptococci was linked to that of *C. matruchotii* and that these mechanisms support the previously described "corncob" arrangement between these species but that this is favorable only in aerobic conditions. Furthermore, we utilized scanning electrochemical microscopy to quantify lactate production and consumption between individual bacterial cells for the first time, revealing that lactate oxidation provides a fitness benefit to *S. mitis* not due to pH mitigation. This study describes mechanistic interactions between two highly abundant human commensals that can explain their observed *in vivo* spatial arrangements and suggest a way by which they may help preserve a healthy oral bacterial community.

**IMPORTANCE**    As the microbiome era matures, the need for mechanistic interaction data between species is crucial to understand how stable microbiomes are preserved, especially in healthy conditions where the microbiota could help resist opportunistic or exogenous pathogens. Here we reveal multiple mechanisms of interaction between two commensals that dictate their biogeographic relationship to each other in previously described structures in human supragingival plaque. Using a novel variation for chemical detection, we observed metabolite exchange between individual bacterial cells in real time validating the ability of these organisms to carry out metabolic crossfeeding at distal and temporal scales observed *in vivo*. These findings reveal one way by which these interactions are both favorable to the interacting commensals and potentially the host.

**KEYWORDS**    single cell, polymicrobial, microbiome, commensal

Over the past decades, our knowledge of the human oral microbiome has increased drastically, revealing a robust polymicrobial biofilm in supragingival plaque (SUPP) that is present in healthy as well as in diseased conditions. While we know a great deal about what bacteria reside in SUPP (1–4), we know very little about the interactions between taxa especially in healthy conditions relative to disease. Given that

Address correspondence to Matthew Ramsey, mramsey@uri.edu.

Eric Almeida and Surendra Puri contributed equally to this article. Co-authors are listed in surname alphabetical order.

The authors declare no conflict of interest.

See the funding table on p. 12.

dysbiosis of the healthy microbiota is often a prelude to oral disease (5–7), we wish to study interactions within the healthy community to potentially reveal any community members that might help preserve stable community structure and constituency, potentially preventing the onset of disease.

Previous studies have shown the importance of attachment to the development of the oral biofilm (8) and new data have identified and refined the spatial organization/"biogeography" of abundant commensal organisms found in SUPP (9). Human microbiome project data and recent microscopy of healthy individuals have revealed that one of the most abundant and prevalent species in SUPP is *Corynebacterium matruchotii* (1, 4, 9). *C. matruchotii* has been correlated with good dental health and hypothesized to be important in the organization of some plaque biofilm structures particularly due to its ability to adhere to *Streptococcus* species forming a structure referred to as a "corncob" where the *Corynebacterium* filament is surrounded by streptococci (9), a role typically ascribed to *Fusobacterium* (8, 10). *C. matruchotii* has also been shown to help facilitate dental calculus formation via calcification (11). Previous studies have not likely appreciated the role *C. matruchotii* plays in bridging early and late colonizers within the plaque (8, 10) and its importance in the structuring of the plaque community as it is presumed to bind to an existing biofilm of *Streptococcus* and *Actinomyces* cells (9, 12). The spatial organization of microbes in these SUPP biofilm structures has been characterized in the "hedgehog" model (9) which visualizes *C. matruchotii* and its proximity to adjacent *Streptococcus* species, such as *S. mitis* at the SUPP perimeter (9, 13).

*Streptococcus* species, such as *S. mitis*, are one of the most abundant species in the oral microbiome (2) and are known for their ability to compete in their environment by producing antimicrobial metabolites like hydrogen peroxide (14) via the *spxB* gene product, pyruvate oxidase (15). Fermentation by streptococci (primarily lactate production) can decrease local pH which then selects for other microbes including *S. mutans* which thrive in the community causing caries (16–18). Streptococci can also co-aggregate with other species to benefit from their catalase activity (19) and we have previously shown that crossfeeding on *Streptococcus*-produced lactate by adjacent microbes increases their growth yields (20) and that co-proximity results in a catalase-dependent removal of $H_2O_2$ (21). *C. matruchotii*, in close proximity with *S. mitis*, must survive in the presence of these same metabolites and how it does so is unknown. If their interaction were to result in *Corynebacterium*-mediated detoxification of streptococcal metabolites, then this could help stabilize a diverse bacterial biofilm community which may, in turn, enhance colonization resistance (i.e., the ability of these biofilms to limit the growth of opportunistic or exogenous pathogens).

We employed a reductionist approach to investigate the relationship between *C. matruchotii* and *S. mitis*. We discovered that *S. mitis* has a considerable increase in growth yield with *C. matruchotii* aerobically but not anaerobically where *C. matruchotii* growth is also inhibited by *S. mitis*. We also observed that *C. matruchotii* upregulated lactate catabolism genes in coculture with *S. mitis* and surprisingly observed that oxidation of lactate by *C. matruchotii* was a contributor to *S. mitis* growth enhancement and was pH independent. We then utilized scanning electrochemical microscopy (SECM) to demonstrate that *C. matruchotii* lactate catabolism can deplete local concentrations of this organic acid swiftly in real time at submicron scales, implying that metabolite consumption in coculture can occur in the observed *in vivo* arrangements between these organisms. These data reveal mechanisms of interaction that support the *in vivo* co-occurrence and biogeography between these species in healthy oral biofilms.

## RESULTS

Using a reductionist approach, we utilized *in vitro* methods to identify and test hypothetical mechanisms of interaction between *C. matruchotii* and *S. mitis* which live in direct proximity in the SUPP biofilm *in vivo*.

## *C. matruchotii* enhances the growth of *S. mitis* in aerobic conditions

We performed pairwise coculture experiments aerobically and anaerobically with a solid medium colony biofilm model (22) to quantify growth yield between mono- and cocultures of *S. mitis* with *C. matruchotii* (Fig. 1) observing a 954-fold growth yield enhancement of *S. mitis* in coculture. Unexpectedly, *C. matruchotii* had no significant difference in growth yield with *S. mitis* (Fig. 1A). While previous studies have hypothesized that *C. matruchotii*–*Streptococcus* interactions occur in aerobic microenvironments within SUPP (9, 13), we also performed the same experiment in anaerobic conditions as a comparison (Fig. 1B). Interestingly, the coculture growth benefit for *S. mitis* was lost while *C. matruchotii* yield decreased ~130-fold.

## *C. matruchotii* upregulates genes necessary for L-lactate catabolism and oxidative stress response

To investigate how *C. matruchotii* enhances *S. mitis* growth yield in coculture, we performed RNAseq to compare mono- vs coculture transcriptome data. *C. matruchotii* differentially expressed only 22 genes (greater than two-fold) in aerobic coculture with *S. mitis* (Table S1). Interestingly, *C. matruchotii* upregulated the *lutABC* operon (*lutA*, 4.37-fold; *lutB*, 3.76-fold; and *lutC*, 3.20-fold), whose gene products in *Bacillus subtilis* catabolize L-lactate (23) using oxygen as a terminal electron acceptor. Thus, in the absence of oxygen, *C. matruchotii* is no longer able to catabolize L-lactate, as previously shown (24). *C. matruchotii* also significantly upregulates a bacterial non-heme ferritin-encoding gene (2.39-fold) in coculture. This protein has been characterized in *Mycobacterium smegmatis* to sequester ferrous ions as part of the oxidative stress response (25). Given the coculture growth and transcriptome results, we broadly hypothesized that *C. matruchotii* crossfeeds on *S. mitis*-produced lactate while detoxifying *S. mitis*-produced $H_2O_2$ similar to other microbes in the oral cavity (19, 20). Given the fact that *C. matruchotii* cannot utilize L-lactate anaerobically and *S. mitis* only provided a growth benefit in the presence of oxygen, we believe these data suggest one mechanism by which the biogeography of these species *in vivo* could be influenced by their metabolic interactions.

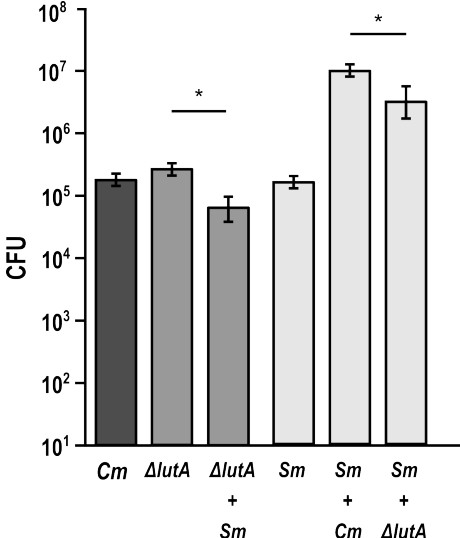

**FIG 1** Growth yield measurements of mono- vs coculture biofilms. Aerobic (A) and anaerobic (B) colony forming unit (CFU) counts of *C. matruchotii* (*Cm*) and *S. mitis* (*Sm*) in mono and cocultures. Data are mean CFU counts for *n* ≥ 3 and error bars represent 1 standard deviation. * denotes *P* < 0.05 using a Student *t*-test.

## Lactate utilization by *C. matruchotii* influences *S. mitis* growth yield

The growth enhancement of *S. mitis* in coculture with *C. matruchotii* is likely due to several factors, including $H_2O_2$ decomposition and lactate catabolism. It is unclear whether the removal of lactate itself or the removal of lactate and subsequent increase in pH is responsible for *S. mitis* growth yield enhancement. We first tested the impact of pH by performing growth experiments in the same medium with increased buffer capacity by adding 50 mM 3-(N-morpholino)propanesulfonic acid (MOPS). Qualitatively, we observed that *S. mitis* monoculture colonies no longer produced yellow coloration in buffered medium containing the pH indicator dye phenol red (i.e., no longer acidified the environment) compared to the original medium (data not shown). Quantitatively, we observed that *S. mitis* growth yield had no significant change in monoculture with additional MOPS (Fig. S1), indicating that pH was not responsible for *S. mitis* growth yield increases in coculture.

To determine whether the removal of lactate by *C. matruchotii* via catabolism was enhancing streptococcal fitness, we constructed a *lutABC* operon deletion mutant (*ΔlutABC*) since each gene within the *lutABC* operon had been described to be essential for L-lactate catabolism (23). The *ΔlutABC* strain was significantly impaired in L-lactate utilization showing a diminished growth rate (doubling times of 12.85 h for the wild type [WT], 50.07 h for *ΔlutABC,* and 13.45 h for the *ΔlutABC* strain with *lutABC* complemented *in trans*) aerobically with L-lactate as the sole carbon source (Table 1). *C. matruchotii* possesses two additional annotated L-lactate dehydrogenases which may function bidirectionally allowing it to slowly oxidize L-lactate without a functional *lutABC* system. A *lutA* deletion strain was also created and showed similar results (Table 1).

We next tested the *ΔlutABC* mutant in mono- vs coculture with *S. mitis* to determine whether impaired lactate utilization led to a decrease in *S. mitis* yield in coculture with *C. matruchotii*. Using defined medium in glucose-limited conditions to force competition for limited glucose and/or promote crossfeeding on streptococcal-produced lactate, we observed that both *S. mitis* and *C. matruchotii ΔlutABC* fitness were significantly decreased in coculture (Fig. 2). *Corynebacterium ΔlutABC* can only poorly catabolize L-lactate (Table 1) and thus poorly crossfeed on *S. mitis*-produced L-lactate compared to the WT. As *ΔlutABC* and *S. mitis* are now forced to compete for limited glucose, both exhibit a decreased growth yield. This is in agreement with previous anaerobic data (Fig. 1B), where lactate oxidation by *C. matruchotii* does not occur. The growth yield increase in *S. mitis* in coculture is diminished when *C. matruchotii* cannot oxidize lactate, but this does not fully explain the total growth benefit provided, suggesting another mechanism(s) at work.

## Catalase abundance leads to enhanced streptococcal growth yields

Given that lactate oxidation by *C. matruchotii* provides only a portion of the fitness benefit in coculture to *S. mitis,* we next investigated whether $H_2O_2$ detoxification by *C. matruchotii* also contributed to the fitness benefit. Surprisingly, in coculture with *S. mitis*, *C. matruchotii* did not upregulate the expression of the single catalase (*katA*) encoded on its chromosome (Table S2). We observed that catalase was maximally expressed aerobically and not expressed anaerobically (data not shown). To test whether catalase-dependent $H_2O_2$ detoxification was important for *C. matruchotii* fitness in coculture and

**TABLE 1** Doubling times for *C. matruchotii, C. matruchotii ΔlutA, ΔlutABC* mutants, and *ΔlutABC* complemented *in trans* with *lutABC*[a]

| Strain | Doubling time | *t*-test vs WT |
|---|---|---|
| *C. matruchotii* WT | 12.83 h ± 1.3 h | |
| *C. matruchotii ΔlutABC* | 58.78 h ± 19.0 h | $1.3 \times 10^{-5}$ |
| *C. matruchotii ΔlutABC* + empty vector (pCGL0243) | 50.07 h ± 3.5 h | $4.3 \times 10^{-6}$ |
| *C. matruchotii ΔlutABC* + *lutABC* complement vector | 13.45 h ± 9.8 h | $1.6 \times 10^{-1}$ |
| *C. matruchotii ΔlutA* | 32.06 h ± 1.8 h | $7.5 \times 10^{-6}$ |

[a]$n = 3$, Student's *t*-test was performed for each group vs the WT.

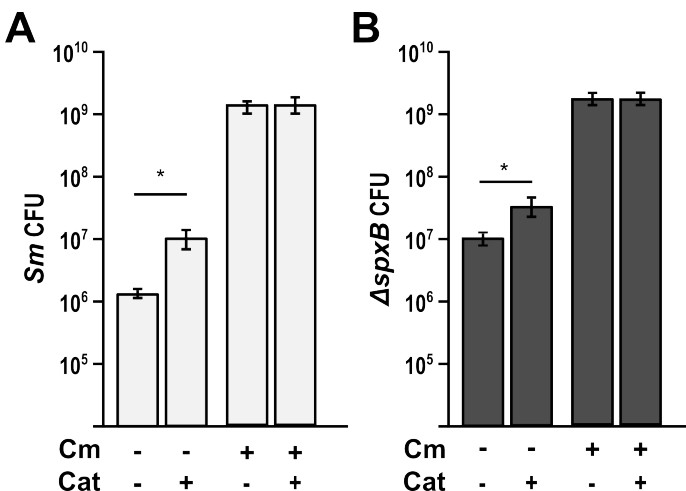

**FIG 2** Limiting glucose colony aerobic biofilm cocultures. Colony forming unit (CFU) counts of *C. matruchotii* (*Cm*), *C. matruchotii ΔlutA*, and *S. mitis* (*Sm*) in mono- and cocultures. Data are mean CFU counts for n ≥ 3 and error bars represent 1 standard deviation. * denotes P < 0.05 using a Student t-test.

subsequent *S. mitis* growth yield enhancement, we generated the catalase gene deletion mutant, *C. matruchotii ΔkatA*. Interestingly, this mutant had to be generated entirely under anaerobic conditions and does not survive incubation in aerobic or microaerophilic conditions, making it impossible to test this mutant in aerobic coculture with *S. mitis*. Instead, we determined the contribution of catalase to the growth of these species in medium amended with 100 U/mL of bovine catalase. Previous studies (19, 26, 27) have indicated that streptococcal-produced $H_2O_2$ is capable of limiting their own growth. We observed that adding exogenous catalase elevated the monoculture growth yield of *S. mitis* 6.42-fold (Fig. 3). This self-limitation by $H_2O_2$ production was also observed when comparing monoculture fold changes in *S. mitis* to the non-$H_2O_2$-producing *ΔspxB* mutant (Fig. 3). Interestingly, the growth benefit of *S. mitis* in coculture with *C. matruchotii* dropped from 954-fold to 148-fold when amended with catalase.

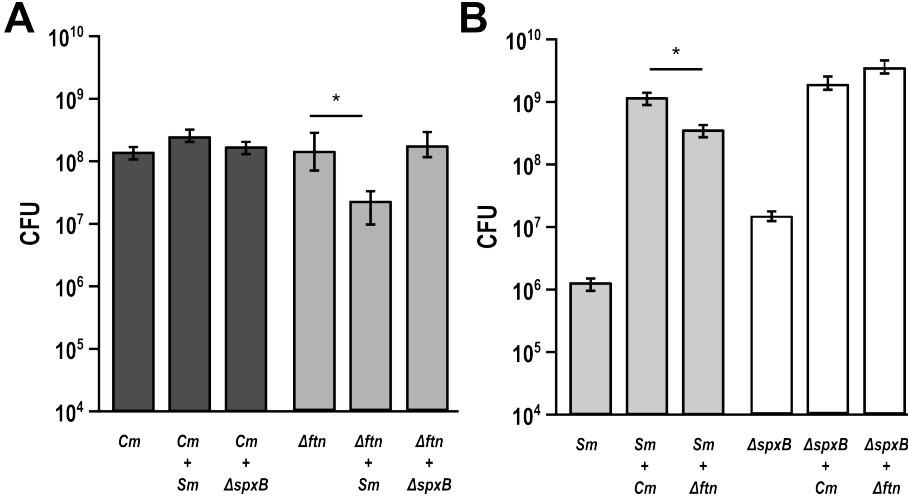

**FIG 3** *S. mitis* aerobic monoculture enhanced by exogenous catalase. (A)Colony forming unit (CFU) counts of *S. mitis* WT (*Sm*) in mono- and coculture with *C. matruchotii* (*Cm*) on media containing 100 U/mL of catalase vs without. (B) CFU counts of *S. mitis ΔspxB* in mono and coculture with *Cm* on media containing 100 U/mL of catalase vs without. * denotes P < 0.05 using a Student t-test.

## C. matruchotii requires a functional oxidative stress response to be fit to interact with S. mitis

In coculture with *S. mitis*, *C matruchotii* significantly upregulated a gene encoding ferritin, a bacterial non-heme protein involved in oxidative stress response (25). We hypothesized that ferritin was needed for *C. matruchotii* fitness with $H_2O_2$-producing streptococci. To test this hypothesis, we deleted the ferritin encoding gene generating *C. matruchotii Δftn* and performed cocultures with WT *S. mitis* and *S. mitis ΔspxB* (which is unable to produce $H_2O_2$) (14). In coculture with WT *S. mitis*, we observed that the *Δftn* mutant fitness decreased 7.35-fold (Fig. 4A) and this decrease was not observed in coculture with the *S. mitis ΔspxB* strain. *S. mitis* WT had a 4.6-fold significant decrease in growth yield with *C. matruchotii Δftn* compared to *C. matruchotii* WT, whereas there was no change in growth yield with *S. mitis ΔspxB* with either *C. matruchotii* strain. This shows that *C. matruchotii* needs a functional oxidative stress response to be fit in interactions with WT *S. mitis*. These data indicate that $H_2O_2$ detoxification is the largest contributor to enhance *S. mitis* fitness in coculture but also that other mechanisms, that is, *C. matruchotii* lactate oxidation, further enhance fitness.

## SECM reveals oxidation of S. mitis-produced lactate by adjacent C. matruchotii at submicron scale

To investigate lactate production and consumption *in situ* by bacteria as well as the topography of bacterial cells, a submicropipette-supported interface between two immiscible electrolyte solutions was employed (Fig. S2A) (28). The full methodology and findings of this work are part of a co-submitted manuscript (29). With this submicrotip, an etched Ni/Cu electrode in the internal organic electrolyte exerts a bias across the submicroscale liquid–liquid interface against an electrode in the aqueous solution to yield the amperometric tip current based on the selective interfacial transfer of a small probe ion (28). The coculture of *C. matruchotii* and *S. mitis* was immobilized over a poly L-lysine-coated glass slide and studied by scanning or approaching an 800-nm-diameter pipette tip over the bacteria. Further details are provided in the supplemental materials (Fig. S2C and D).

### Imaging of cells by volume

We employed the constant height mode of submicronscale SECM to successfully image single bacterial cells in coculture (Fig. 5). The high spatial resolution was obtained using submicropipette tips, which were characterized by cyclic voltammetry for tetraethylammonium (TEA+) ion transfer (IT) *in situ* to obtain a diffusion limited current in the bulk solution, $i_{T,\infty}$ (Fig. S2B). Constant height imaging of cocultured bacteria for the probe ion TEA was obtained with the gap between the tip and the bacteria, $d_C = 0.75$ μm (i.e.,

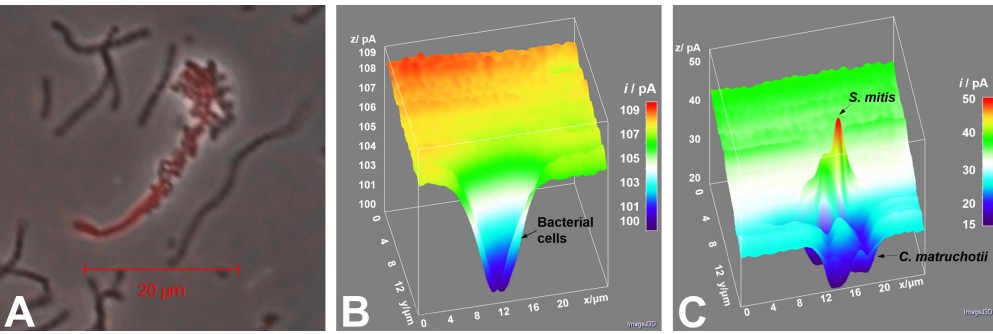

FIG 4   *C. matruchotii* ferritin knockout inhibited when cocultured with *S. mitis*. (A) Aerobic colony forming unit (CFU) counts of *C. matruchotii* WT (*Cm*) and ferritin knockout (*Δftn*) in mono- and coculture with *S. mitis* WT and strain lacking the ability to create $H_2O_2$ (*ΔspxB*). (B) CFU counts of *Sm* and *ΔspxB* in mono and coculture with *Cm* and *Δftn*. Data are mean CFU counts with error bars indicating standard deviation for $n \geq 3$. * denotes $P < 0.05$ using a Student *t*-test compared to monoculture.

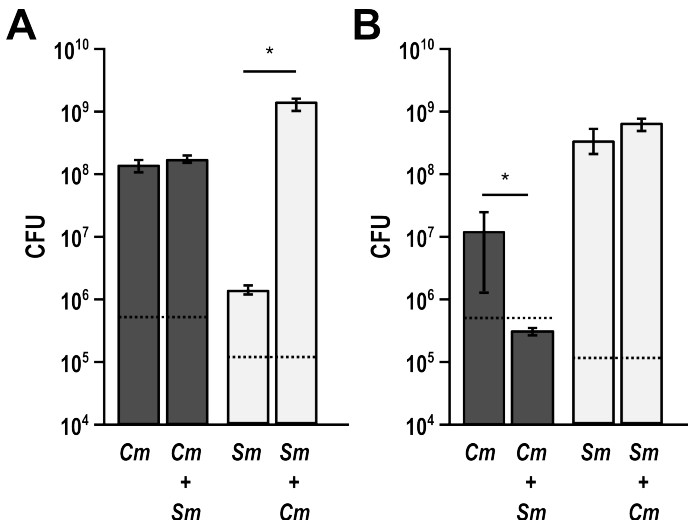

**FIG 5** SECM detection of *C. matruchotii* oxidation of *S. mitis*-produced lactate. (A) Representative micrograph of *C. matruchotii* (red filament) and *S. mitis* (chained spheres) coculture. (B) Topographical SECM image based on TEA$^+$ IT (obtained with a gap between the tip and the bacteria, $d_c$ = 1.8 $d/a$). Monotonic current decrease over bacterial "lump" is observed due to hindered diffusion of TEA$^+$ through the bacterial membrane. (C) Lactate-mapping SECM image based on lactate IT (obtained at $d_c$ = 1.2 $d/a$). Sharp current transition between *S. mitis* and *C. matruchotii* is observed corresponding to simultaneous lactate production and consumption, respectively, with an estimated range of 0.5 mM at the *S. mitis* surface to 0.1 mM above *C. matruchotii*. SECM images were obtained in constant height mode.

1.80 normalized distance to tip radius [$d/a$]). This SECM image could not resolve each individual bacterial cell. For instance, a lump was identified in 25 µm × 20 µm image based on TEA$^+$ IT, which did not resolve any difference between bacterial cells (Fig. 5B). Low tip current of ~80% of $i_{T,\infty}$ for TEA$^+$ above these bacteria was obtained due to hindered diffusion of TEA$^+$ by adjacent bacteria with partially permeable membranes to this probe ion.

### Imaging of lactate gradients around cells

The same area in 25 µm × 20 µm was imaged based on lactate IT with the gap between the tip and the bacteria, $d_c$ = 0.50 µm (1.20 $d/a$), which could resolve individual *S. mitis* and *C. matruchotii* clearly (Fig. 5C). An initial current of 30 pA above a glass substrate corresponds to c.a. 0.26 mM of lactate produced by ensemble of *S. mitis* and diffused to bulk solution near bacteria according to equation 1 below.

$$i_{T,\infty} = 4xzFDCa \qquad (1)$$

where $i_{T,\infty}$ is a current in bulk, $x$ is the function of RG ratio (RG is the ratio of outer and inner diameters of a glass pipette, $x$ = 1.16 for a RG 1.5 tip), $z$ is the charge of lactate, $F$ is Faraday constant (96,485 C/mol), $D$ is the diffusion coefficient (6 × 10$^{-6}$ cm$^2$/s), $C$ is a concentration of lactate (0.26 mM), and $a$ is the inner radius of a pipette tip (430 nm).

In this SECM image, higher tip currents than $i_{T,\infty}$ for lactate are obtained above spherical *S. mitis*, while lower tip currents than $i_{T,\infty}$ are observed above filamentous *C. matruchotii*. Not only distinctive morphologies are clearly distinguished between two different bacteria as shown in an optical microscopic image (Fig. 5A), but also the production and consumption of lactate between them are visually confirmed *in situ*, where currents were dramatically transposed from an enhanced response over *S. mitis* to reduced ones over *C. matruchotii*, implying local increase in lactate produced by *S. mitis* and local depletion of lactate consumed by *C. matruchotii*. Notably, this SECM image successfully visualized the chemical interaction between two commensal oral microbes

in real time and is the first SECM study to our knowledge that measures metabolite exchange between two individual bacterial cells. Specifically, *S. mitis* produces c.a. 0.50 mM lactate locally, which is efficiently depleted by C. *matruchotii* (Fig. 5C; Fig. S2E), thus verifying a standing question about their commensal relationship that cannot be answered only by optical microscopic imaging. Quantitative analysis of the permeability of the bacterial membrane and the local concentration of lactate produced by *S. mitis* are discussed in the supplemental materials and companion manuscript detailing this methodology (29).

## DISCUSSION

Interactions between commensal bacteria in the human microbiome are quite under-studied. This is especially the case within healthy SUPP where they likely have a role in maintaining plaque homeostasis and host health compared to subgingival plaque and oral disease (8, 10, 30). While the organisms in plaque biofilms are in close proximity and capable of physical and biochemical interaction, the involved mechanisms are largely hypothetical (9). Characterizing the behavior of abundant SUPP commensal organisms can help reveal necessary interactions that could maintain a healthy microbiome as well as explain their biogeographic arrangements. One set of interactions is those between *Corynebacterium matruchotii* and *Streptococcus* spp. in previously described "hedgehog" structures (9, 13), where they occur at the presumed aerobic biofilm/saliva margin. This study investigates these interactions and provides novel data on metabolite exchange between individual cells that have broad implications on polymicrobial biofilms beyond the human oral cavity.

The importance of $H_2O_2$ production (via SpxB (31)) by oral streptococci has been intensively studied by many groups (32, 33). The ability of these species to generate $H_2O_2$ has been shown to modulate host immune responses (34) as well as compete with adjacent microbes including other streptococci, *Porphyromonas gingivalis,* and many others (35–37). In addition, the ability of oral streptococci to acidify their environment via lactate production is well understood, particularly for *S. mutans* and the ability of adjacent organisms to tolerate acid stress via neutralization or consumption of lactate is well documented (20, 24, 38). Lactate metabolism particularly defines the relationship between fermenting *Streptococcus* spp. and the lactate-oxidizing *Veillonella* spp. (39). We have also previously studied the impact of adjacent microbes on the mitigation of $H_2O_2$ via SECM (21). These findings led us to make similar hypotheses for $H_2O_2$ and lactate metabolism regarding C. *matruchotii*–*Streptococcus* interactions and their influence on the fitness of either species. Our data indicate that *S. mitis* had a significant growth yield increase when cocultured with C. *matruchotii* (Fig. 1A) and this growth benefit was lost anaerobically (Fig. 1B) which can partly explain their proximal association only at the aerobic margin of hedgehog structures (9, 13). $H_2O_2$-producing *Streptococcus* and adjacent commensal species have been shown to coexist despite reactive oxygen species (ROS) production (19). C. *matruchotii* is uninhibited when cocultured with *S. mitis* aerobically likely due to its catalase production. While a C. *matruchotii* Δ*katA* mutant could not grow aerobically, we were able to demonstrate that the addition of exogenous catalase to WT monoculture resulted in enhancing *S. mitis* yield like that of the non-peroxide-producing *S. mitis* spxB mutant but not to the same extent observed in coculture with C. *matruchotii* with or without exogenous catalase (Fig. 3). These data suggest that catalase can enhance *S. mitis* growth, even if produced by adjacent C. *matruchotii*, similar to observations we have made previously (21), but also suggest that additional benefits to *S. mitis* exist due to coculture. Interestingly, we observed that coculture led to upregulation in C. *matruchotii* of a gene encoding a protein that has 82% identity to ferritin from C. *mustelae*. Ferritins have been shown to protect from ROS by sequestering iron and binding to DNA (25) to prevent the production of hydroxyl radicals (40). A C. *matruchotii* Δ*ftn* mutant showed a significant yield decrease in coculture with *S. mitis,* but decreases were not observed with *S. mitis* Δ*spxB* (Fig. 4). We observed that any decreases in C. *matruchotii* yield were mirrored by decreases in *S. mitis* yield as well,

linking streptococcal fitness to that of *C. matruchotii*. We hypothesize that this should also be true for any other adjacent $H_2O_2$-producing streptococcal species. Transcriptional responses of *S. mitis* to *C. matruchotii* are part of a separate ongoing study and are not reflected here.

Of the 22 genes that *C. matruchotii* differentially expresses with *S. mitis* aerobically, three belong to the lactate oxidization encoding *lutABC* operon (23). *C. matruchotii* can only oxidize lactate aerobically and will not oxidize lactate anaerobically (24) which can explain its inability to grow with *S. mitis* in anerobic coculture (Fig. 1B). Initially, we hypothesized that *C. matruchotii* oxidation of streptococci-produced lactate could provide a growth benefit to adjacent streptococci by elevating the local pH. However, no amount of additional MOPS buffer was sufficient to increase *S. mitis* monoculture yields (Fig. S1), despite local acidification no longer being detectable via a phenol red indicator dye. Thus, we hypothesized that the removal of lactate itself (and elimination of feedback inhibition of this fermentation product) would benefit *S. mitis* in a pH-independent fashion by allowing it to ferment more carbohydrates. To test this, we deleted the *lut* operon in *C. matruchotii* and competed this strain in a glucose-limited coculture where *C. matruchotii* should rely on crossfeeding of streptococci-produced lactate. *C. matruchotii* *ΔlutABC* yields were significantly decreased in coculture with *S. mitis* in limiting glucose when compared to monoculture with a similar decrease in *S. mitis* yield (Fig. 2). While these changes were significant, they were modest, which is due to the ability of *C. matruchotii* *ΔlutABC* strains to still oxidize lactate to a minor extent, likely due to the presence of at least two potentially reversible lactate dehydrogenase encoding genes (Fig. S2). Even with only partial loss of function, these results indicate that *C. matruchotii* cannot compete for glucose with *S. mitis* and likely depends on crossfeeding of lactate when they are in direct proximity.

Using SECM, we were able to directly quantify lactate production by *S. mitis* and its oxidation by adjacent *C. matruchotii* in real time (Fig. 5) (29), indicating a sharp decrease in lactate concentration between individual cells. To the best of our knowledge, this is the first observation of metabolite exchange between individual bacterial cells by SECM. Based on these data, we believe that existing "corncob" configurations observed *in vivo* (9, 13) should readily be able to consistently remove lactate from their immediate area. Interestingly, we also observed in our companion study (29) that lactate utilization in an individual *C. matruchotii* filament was localized nearest to streptococcal cells which also fits well with their observed *in vivo* arrangement toward the aerobic-oriented pole of the corncob structure (9, 13). This would allow streptococcal metabolism to continue without inhibition while eliminating a source of acid stress to the host and other adjacent microbiota which contrasts well with the dense clustering biogeography of *S. mutans* observed on enamel during caries formation (17). This observation supports a mechanism, whereby the interaction between these two commensals may contribute to the lack of cariogenic activity in a healthy oral biofilm. Previous studies by Frenkel and Ribbeck (41, 42) have demonstrated that physical separation of streptococcal aggregates via mucins is sufficient to enhance the growth of competing species and limit damage to enamel. This is reminiscent of distal separation of streptococci bound to *C. matruchotii* "corncobs"; yet, these can be further enhanced by their ability to also remove lactate actively from the coculture environment with spatial heterogeneity that favors the observed biogeographic heterogeneity (9, 13).

This study has described two mechanisms of interaction between bacteria that exist in direct contact *in vivo*. Using a reductionist approach, we were able to ascertain how each mechanism contributed to fitness of both organisms. Advantages provided to each species when these mechanisms are intact also explain the positional/biogeographic arrangement of these species *in vivo*. In addition, we were able to demonstrate real-time metabolite exchange between these species at submicron distances, indicating that crossfeeding between these organisms is likely occurring between them *in vivo*. These interactions reveal how structural orientation and species composition between commensals may contribute to host health and potentially be one way by which a

healthy biofilm composition is maintained *in vivo* and answer the hypotheses about mechanisms of interaction between these organisms that were hypothesized several years earlier (9).

## MATERIALS AND METHODS

### Strains and media

Strains and plasmids used in this study are listed in Table S1. *C. matruchotii* (ATCC 14266) and *S. mitis* (ATCC 49456) were grown on the Brain Heart Infusion (BHI) media supplemented with 0.5% of yeast extract (YE) at 37°C in a static incubator with 5% $CO_2$ or in 5% $H_2$, 10% $CO_2$, and 85% $N_2$ in anaerobic conditions. *Escherichia coli* was grown at 37°C in standard atmospheric conditions with liquid cultures shaken at 200 rpm. Antibiotics were used at the following concentrations: kanamycin 40 µg/mL for *E. coli* and 10 µg/mL for *C. matruchotii*.

### Colony biofilm coculture/buffered coculture/ catalase coculture

Overnight cultures of *C. matruchotii* and *S. mitis* species were grown in BHI media supplemented with 0.5% of YE aerobically and anaerobically as described above. Colony biofilm assays were carried out as described previously (22). Briefly, a semipermeable 0.22-µm polycarbonate membrane filter (43) was placed on solid BHI-YE media (supplemented with 1.6% agar). Ten microliters of each culture was spotted on the membrane filters and monocultures were spot with 10 µL of BHI-YE. The cultures were incubated for 48 h and the membranes were placed in a microcentrifuge tube with 1 mL of BHI-YE. The tubes were vortexed to resuspend into media and serially diluted and track plated (44) to count colony forming units per mL (CFU/mL). *S. mitis* was counted using BHI-YE plates and *C. matruchotii* on BHI-YE plates supplemented with 100 µg/mL of fosfomycin. Buffered and pH indicator cocultures were carried out with 50 mM MOPS and 18 mg/mL of phenol red added to BHI-YE. Catalase cocultures were carried out with 100 U/mL of catalase added to BHI-YE.

### RNAseq experiment and analysis

Mono and cocultures were prepared similar to the colony biofilm coculture with the exception that culture membranes were incubated for 24 h and moved to fresh media for an additional 4 h. Membranes were then removed from solid agar and immediately placed into RNALater (Ambion), where cells were removed by agitation and pelleted by centrifugation. Cell pellets were stored in the Trizol reagent at −80°C. Experiments were carried out in biological duplicates. RNA extraction, library preparation, and sequencing were then carried out by the Microbial 'Omics Core facility at the Broad Institute. RNASeq libraries were generated using previously described methods (45). Sequence data were aligned using Bowtie2 (46) and read counts per coding sequence were called using HTSeq-Count (47). Statistical analysis was carried out via DESeq2 (29) to determine differentially expressed genes. Scripts of this pipeline can be found at https://github.com/dasithperera-hub/RNASeq-analysis-toolkit. Sequence libraries are available through the NCBI short read archive under bioproject number PRJNA832032.

### Gene deletions

All *C. matruchotii* gene deletions were carried out with sucrose counterselection using a suicide vector derived from pMRKO (20), pEAKO2 which contains *sacB* from pK19mob-sacB (48). Approximately 1,000 bp upstream and downstream flanking regions for each gene were used for homologous recombination and fragments were cloned into pEAKO2 via the Gibson Assembly (49). *C. matruchotii* cells were made as previously described (50). Transformations were carried out with 50 µL of competent cells and 1 µg of DNA electroporated with 0.2-cm gap cuvettes at a voltage of 2.5 kV, a resistance of 400 Ω, and a capacitance of 25 µF. After electroporation, 950 mL of prewarmed BHI-YE was added

to the cuvette and the mixture was then moved to a 46°C heat block for 6 min. After heat shock, transformations were shaken at 250 rpm at 37°C for 4 h. Transformations were plated on BHI-YE Kan$_{10}$ plates and incubated for 4 d at 37°C. Mutants were verified through PCR.

## Gene complementation

The *lutABC* coding sequence and its upstream native promoter region were amplified using primers with overlaps designed for Gibson Assembly (via https://neb-uilder.neb.com/#!/). This 3,586 bp amplicon was cloned into the BamHI-digested pCGL0243 *Corynebacterium* shuttle vector (51) using identical methods to gene deletion constructs detailed above. The *lutABC* complement vector and empty pCGL0243 vectors were, respectively, electroporated into the *C. matruchotii lutABC* gene deletion strain and selected for growth on kanamycin. Successful vector assembly was confirmed by restriction digest mapping.

## Limiting glucose coculture

Cultures were prepared like colony biofilm coculture described above except for being inoculated into 2 mL of liquid-defined medium. Modified RPMI medium (Gibco) was used as a base and supplemented with 8 mM glucose. Cocultures were inoculated for 48 h and track plated to determine CFU/mL.

## SECM sample preparation

Bacterial sample preparation was performed using a previously described defined medium (52) and glucose at 10 mM. Further details are provided in the Supplemental Materials.

## SECM acquisition

Scanning parameters and nano-probe design are similar to methods described previously (53–57). A full description of SECM calibration, sample acquisition, and metabolite quantification is provided in the Supplemental Materials and in a companion manuscript (48).

## ACKNOWLEDGMENTS

We thank Janet Atoyan and the RI-EPSCOR sequencing facility at URI for sequence generation; Jonathan Livny and the Microbial 'Omics Core and Genomics Platform for their help with RNASeq library sequencing and guidance on experimental design; and other members of the Ramsey lab and the Annual Mark Wilson conference attendees for many valuable suggestions and discussion.

This work was funded by the NIDCR/NIH (R01DE027958, M.R.), NIGMS/RI-INBRE early career development award (P20GM103430, M.R.), and the USDA National Institute of Food and Agriculture, Hatch Formula project accession number 1017848 (M.R.). Also, electrochemical study including SECM measurements was partially supported by the National Science Foundation (CHE-2046363).

M.R. and J.K. designed the research; E.A., S.P., and S.E. performed the research; and E.A., S.P., J.K., and M.R. wrote the paper.

There are no competing interests.

## AUTHOR AFFILIATIONS

[1]Department of Cell and Molecular Biology, The University of Rhode Island, Kingston, Rhode Island, USA
[2]Department of Chemistry, The University of Rhode Island, Kingston, Rhode Island, USA

## AUTHOR ORCIDs

Matthew Ramsey  http://orcid.org/0000-0001-7158-5120

## FUNDING

| Funder | Grant(s) | Author(s) |
|---|---|---|
| HHS | NIH | National Institute of Dental and Craniofacial Research (NIDCR) | R01DE027958 | Matthew Mark Ramsey |
| | | Eric Almeida |
| HHS | NIH | National Institute of General Medical Sciences (NIGMS) | P20GM103430 | Matthew Mark Ramsey |
| | | Eric Almeida |
| U.S. Department of Agriculture (USDA) | 1017848 | Matthew Mark Ramsey |
| | | Eric Almeida |
| National Science Foundation (NSF) | CHE-2046363 | Jiyeon Kim |
| | | Surendra Puri |
| | | Subashini Elangovan |

## AUTHOR CONTRIBUTIONS

Eric Almeida, Data curation, Formal analysis, Investigation, Methodology, Validation, Visualization, Writing – original draft, Writing – review and editing | Surendra Puri, Data curation, Formal analysis, Investigation, Methodology, Validation, Visualization, Writing – original draft, Writing – review and editing | Alex Labossiere, Formal analysis, Investigation, Writing – review and editing | Subashini Elangovan, Investigation | Jiyeon Kim, Conceptualization, Data curation, Formal analysis, Investigation, Methodology, Project administration, Resources, Supervision, Validation, Visualization, Writing – original draft, Writing – review and editing | Matthew Ramsey, Conceptualization, Data curation, Formal analysis, Funding acquisition, Methodology, Project administration, Resources, Supervision, Validation, Visualization, Writing – original draft, Writing – review and editing

## DATA AVAILABILITY

Scripts utilized for RNASeq analysis can be found at https://github.com/dasithperera-hub/RNASeq-analysis-toolkit. Sequence libraries are available through the NCBI short read archive (SRA) under bioproject number PRJNA832032.

## ADDITIONAL FILES

The following material is available online.

### Supplemental Material

**Supplemental material (mSystems.00115-23-s0001.pdf).** Supplemental results and materials and methods.

### Open Peer Review

**PEER REVIEW HISTORY (review-history.pdf).** An accounting of the reviewer comments and feedback.

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
