## [Reviewer comments · mSystems]

Bacterial multispecies interaction mechanisms dictate biogeographical arrangement between the oral commensals *Corynebacterium matruchotii* and *Streptococcus mitis*

Eric Almeida, Surendra Puri, Alex Labossiere, Subashini Elangovan, Jiyeon Kim, and Matthew Ramsey

Corresponding Author(s): Matthew Ramsey, University of Rhode Island College of the Environment and Life Sciences

Review Timeline:

Submission Date:	February 6, 2023
Editorial Decision:	February 23, 2023
Revision Received:	June 16, 2023
Editorial Decision:	June 19, 2023
Revision Received:	June 27, 2023
Accepted:	June 28, 2023

Editor: Matthew Traxler

Reviewer(s): The reviewers have opted to remain anonymous.

Transaction Report:

DOI: <https://doi.org/10.1128/msystems.00115-23>

February 23, 2023

Dr. Matthew Mark Ramsey
University of Rhode Island
Cell and Molecular Biology
120 Flagg Rd. CBL5 Rm 387
Kingston, RI 02881

Re: mSystems00115-23 (Bacterial multispecies interaction mechanisms dictate biogeographical arrangement between the oral commensals *Corynebacterium matruchotii* and *Streptococcus mitis*)

Dear Dr. Matthew Mark Ramsey:

Thank you for submitting your manuscript to mSystems. We have completed our review and I am pleased to inform you that the decision is potentially accept with 'minor modifications'. However, acceptance will not be final until you have adequately addressed the reviewer comments. I want to draw specific attention to the points raised by Reviewer 2 who notes that complementation of mutants was not conducted. While I do not think a comprehensive redo of all experiments with complemented strains is required, providing evidence that key phenotypes can be complemented would add significant confidence to the findings. Barring this, a compelling reason for why complementations were not conducted would be in order.

Preparing Revision Guidelines

Sincerely,

Matthew Traxler

Editor, mSystems

Journals Department

Reviewer comments:

Reviewer #1 (Comments for the Author):

The manuscript describes a dual species interaction between two prominent and relevant members of the oral biofilm, *S. mitis* and *C. matruchotii*. The authors demonstrate that the metabolic exchange between both species is a benefitting factor in this dual species community. *S. mitis* is a producer of lactate and hydrogen peroxide, and *C. matruchotii* is able to metabolize lactate and detoxify hydrogen peroxide, thus counteracting the self inhibitory effect of hydrogen peroxide on *S. mitis*. Overall this manuscript is well written and highly relevant to oral biofilm development and provides a mechanistic explanation of species distribution. The usage of SECM to demonstrate metabolite exchange is highly innovative.

I only have minor comments:

For Fig. 5, it is not clear how representative this image is. Was this done multiple times? Is there some kind of quantification available?

It would be nice to include a bigger picture in the discussion, including some of the other species that are known to metabolize lactic acid or detoxify hydrogen peroxide.

Supplemental files:

page 4 line 147: should it read probe instead of problem?

Fig. S2...growth curve should be in log-scale

Reviewer #2 (Comments for the Author):

In this manuscript, Almeida and Puri et al. were interesting in characterizing metabolite-mediated interactions between the common oral commensal bacteria *Corynebacterium matruchotii* and *Streptococcus mitis*. To do so, the authors use a combination of traditional bacterial coculture and genetics, as well as a novel method for directly visualizing metabolic exchange between cells that they call scanning electrochemical microscopy (SECM). Together, Almeida and Puri determine that *C. matruchotii* is able to support to growth of *S. mitis* by detoxifying produced hydrogen peroxide and consuming produced lactate. The culture experiments are sound, but I have questions regarding the genetics and SECM methods.

Major Comments

1. The authors generate numerous mutants in both *C. matruchotii* and *S. mitis* throughout this manuscript, but they never complement the mutations in trans to confirm that their deletions are specifically responsible for the effects that they observe.
2. How are the authors sure that their SECM method is specifically measuring L-lactate concentrations? For instance, have the authors repeated their SECM experiment using a *S. mitis* Δ ldh mutant? What about addition of an LDH inhibitor like oxamic acid? Alternatively, what happens if the authors image a coculture of *S. mitis* and a *C. matruchotii* mutant deleted lutABC and the two potentially reversible lactate dehydrogenase genes?
3. Much of the Supplemental Results could be incorporated into the main manuscript text and would better inform readers as to how the SECM method works

Minor Comments

L136: Why were these cutoffs chosen for the RNA seq experiment? Can the authors provide the corresponding P-values in Table S2 and S3?

L161, 174, 196-7: The authors state "data not shown" in several instances.

L242-63: Much of this text belongs in the methods section

L317-9: If the transcriptional responses of *S. mitis* to *C. matruchotii* are part of a separate study, why are they included as Table S3?

Figure 2, 3: Include details in the legends indicating that these experiments were under aerobic conditions

Figure 5: Can the authors report the data in panels B and C with respect to measured L-lactate concentration?

Figure S2: The legend indicates that this experiment was not replicated (i.e., N=1). The authors should replicate this experiment before reporting the results at L170.

Line Comments

L36-7: What is meant by "directly adjacent in vivo"?

L44,157,272: Replace "1st" with "first"

L51: Be more clear about what "they" refers to in this sentence

L77,81: Avoid opening sentences with "It". Be more clear was to what "it" is referring to

L80: replace "with the" with "where the"

L80: replace "being" with "it"

L86-7: What is the "hedgehog" model? Why is it relevant for this work?

L91: Spell out hydrogen peroxide fully during its first use

L93: Sentence has two instances of "which can"

L98, 136, 141, 171, 181, 391, 321, 332, 376, 393, 415: If a species name starts a sentence, then the genus should be written out fully

L100: add "-mediated" after *Corynebacterium*

L102,247-8: Enclose the i.e. statement in parentheses

L105-8: Recommend splitting into two sentences

L122: Would suggest including a preamble sentence before jumping into results

L130-1: Would move this sentence to the start of the following paragraph

L137-41: Would suggest splitting into two sentences

L184: Replace "data anaerobically" with "anaerobic data"

L195: Would suggest referencing Table S2 here

L203: What is "it" referring to?

L217: add "hypothesis" after "to test this..."

L218,219,221,222,224: The "WT" abbreviation has been defined

L310: add "led to" before "upregulation"

L361: What is "(KIM REF+)?"

L384-6: This sentence is repeated from L376-9

P411: Several instances of "will be" in this section

To the editor and reviewers:

Please note that all responses to the reviewers are here in red and original review comments in black. We thank both reviewers for their requests and comments and we appreciate their time in reviewing and helping to improve this manuscript. We outline below our attempts to address all issues raised by writing and further experimentation. We would like to note that our companion manuscript regarding more of the operational details of the SECM technology used here has been accepted and now in press (DOI: 10.1021/acs.analchem.3c01498).

Also, all line numbers used in the response below refer to the original main draft “marked up” copy which is a track changes copy showing where text has been modified in response to reviewer comments. We again thank the reviewers and apologize if we were unable to completely satisfy every request given. We have done our best to explain any circumstances as to why we chose to present some information in the format here.

Reviewer #1 (Comments for the Author):

The manuscript describes a dual species interaction between two prominent and relevant members of the oral biofilm, *S. mitis* and *C. matruchotii*. The authors demonstrate that the metabolic exchange between both species is a benefitting factor in this dual species community. *S. mitis* is a producer of lactate and hydrogen peroxide, and *C. matruchotii* is able to metabolize lactate and detoxify hydrogen peroxide, thus counteracting the self inhibitory effect of hydrogen peroxide on *S. mitis*. Overall this manuscript is well written and highly relevant to oral biofilm development and provides a mechanistic explanation of species distribution. The usage of SECM to demonstrate metabolite exchange is highly innovative.

I only have **minor comments**:

For Fig. 5, it is not clear how representative this image is. Was this done multiple times? Is there some kind of quantification available?

Similar SECM images were reproducibly obtained over >3 coculture samples, where higher current over *S. mitis* and lower current over *C. matruchotii* than the background current indicate in situ production and consumption of lactate, respectively. In the main text (in page 7, line 254~275) and supporting information (in page S4 and Fig S3 in page S7), we mentioned that single *S. mitis* cells produce lactate up to 0.5 mM estimated by current in Fig 5C, theoretically simulated current-distance curves, and theoretically simulated concentration profile in Fig S3E and S3F. At the same time, *C. matruchotii* rapidly consumes lactate produced by *S. mitis* at a rate of $\geq 5 \times 10^6 \text{ s}^{-1}$, thereby decreasing the local concentration of lactate up to 0.1 mM over *C. matruchotii*. More detailed information and rigorous discussion about the theoretical simulation and the quantitative analysis should be found in the companion manuscript now accepted in Analytical Chemistry (10.1021/acs.analchem.3c01498) (submission copies provided to Editor). In the companion manuscript, we proved that only *S. mitis* produces lactate, while *C. matruchotii* doesn't produce any lactate in each monoculture study. Single cells of *S. mitis* in monoculture produce similar level of lactate to coculture sample in below Fig S2B and S2C (companion manuscript). In contrast, a single cell of *C. matruchotii* in monoculture does not produce any lactate, thus nothing appeared in the SECM image based on lactate ion transfer (Fig S3B,

companion manuscript). Herein, we share the additional SECM data (Fig S2 and S3) from the submitted companion manuscript as below for review purposes.

Figure S1. SECM images of *S. mitis* monoculture based on (A) TEA⁺ IT for topography and (B) lactate IT for lactate mapping. A tip scan rate at 100 nm/100 ms during SECM imaging. (C) chronoamperometric responses based on lactate IT (raw data, cross sections of SECM images in (B)). The current polarity is set to negative for anionic current responses.

Figure S2. SECM images of *C. matruchotii* monoculture based on (A) TEA⁺ IT for topography and (B) lactate IT for lactate mapping. A tip scan rate at 100 nm/100 ms during SECM imaging.

It would be nice to include a bigger picture in the discussion, including some of the other species that are known to metabolize lactic acid or detoxify hydrogen peroxide.

We thank the reviewer for this suggestion. Further discussion of peroxide production and lactate metabolism are included in lines 307-319.

Supplemental files:

page 4 line 147: should it read probe instead of problem?

We thank the reviewer for their careful attention!

It should be a two phase “problem” and we have corrected this error. Using COMSOL Multiphysics, we solved a diffusion problem of lactate between two phases of aqueous bulk phase and a single bacterium with laterally homogeneous membrane having a uniform permeability.

Fig. S2...growth curve should be in log-scale

We thank the reviewer for this correction, as part of other reviewer comments these data have now been replaced by table S1 in the supplement.

Reviewer #2 (Comments for the Author):

In this manuscript, Almeida and Puri et al. were interesting in characterizing metabolite-mediated interactions between the common oral commensal bacteria *Corynebacterium matruchotii* and *Streptococcus mitis*. To do so, the authors use a combination of traditional bacterial coculture and genetics, as well as a novel method for directly visualizing metabolic exchange between cells that they call scanning electrochemical microscopy (SCEM). Together, Almeida and Puri determine that *C. matruchotii* is able to support to growth of *S. mitis* by detoxifying produced hydrogen peroxide and consuming produced lactate. The culture experiments are sound, but I have questions regarding the genetics and SCEM methods.

Major Comments

1. The authors generate numerous mutants in both *C. matruchotii* and *S. mitis* throughout this manuscript, but they never complement the mutations *in trans* to confirm that their deletions are specifically responsible for the effects that they observe.

We have now generated and tested an *in trans* complement of *lutABC* to the original Δ *lutABC C. matruchotii* mutant. We verified here that the loss of growth rate in the *lutABC* mutant was complemented by addition of the *lutABC* operon *in trans* vs an empty vector Δ *lutABC C. matruchotii*. See lines 178-9, Materials and Methods and table S1. Also, please note the differences in growth rates from

previous experiments were due to performing the new rate measurements in a different culture vessel type than previously.

The role of SpxB in H₂O₂ generation in *S. mitis* and other species has been previously well described which we discuss and provide further citations in response to reviewer 1 comments on lines 307-319. Likewise, complementation of *spxB* and restoration of H₂O₂ production in *S. mitis* has been previously described (<https://www.nature.com/articles/s41598-021-04562-4#Sec2>). We demonstrate here that removal of peroxide by exogenous catalase yielded a phenotypically similar result to that of the Δ *spxB* *S. mitis* (Fig. 3).

2. How are the authors sure that their SCEM method is specifically measuring L-lactate concentrations?

For instance, have the authors repeated their SCEM experiment using a *S. mitis* Δ ldh mutant? What about addition of an LDH inhibitor like oxamic acid? Alternatively, what happens if the authors image a coculture of *S. mitis* and a *C. matruchotii* mutant deleted *lutABC* and the two potentially reversible lactate dehydrogenase genes?

We independently measured the lactate ion transfer in aqueous bulk solution containing 2 mM lactate in Fig S3B, where lactate ion transfer is induced at \sim 0.42 V more positive than $E_{1/2}$ of TEA⁺ ion transfer. The limiting currents for lactate ion transfer respond to the lactate concentration linearly as well. This bulk electrochemistry confirmed the selective potential for lactate ion transfer across the pipet tip interface and confirms that we are detecting lactate.

Once we defined the potential for lactate ion transfer under nearly diffusion-limited condition in the bulk solution, we applied the same potential to the tip during raster-scanning above co-cultured bacteria. In the companion manuscript now in print at Analytical Chemistry (DOI 10.1021/acs.analchem.3c01498) (and provided to the editor), we demonstrated that only *S. mitis* produces lactate, while *C. matruchotii* does not. Single cells of *S. mitis* in monoculture produce similar level of lactate to coculture sample in Fig S2B and S2C (companion manuscript). In contrast, single cells of *C. matruchotii* in monoculture do not produce any lactate, thus nothing appeared in the SECM image based on lactate ion transfer (Fig S3B, companion manuscript). The relevant companion manuscript figures have been included above in replies to reviewer 1 above.

Regarding an *ldh* mutant given that we had already verified our ability to detect lactate specifically in sterile medium with and without lactate added, we did not see a rationale for inhibiting *ldh* after the fact. Further, given that Streptococcal metabolism is highly dependent on glycolysis and that they do not respire, the likelihood of this mutant surviving much less behaving reproducibly in our SECM settings was unlikely thus it was not attempted. Lastly, we wish to note that we cannot differentiate between L or D lactate with SECM but that *C. matruchotii* does not appear to possess a lactate racemase nor D-lactate dehydrogenase.

3. Much of the Supplemental Results could be incorporated into the main manuscript text and would better inform readers as to how the SCEM method works

We thank the reviewer and agree with this comment. However, given the extensive nature of SECM work included it was already necessary to publish this in a second manuscript given the size constrains

and technical nature of the SECM developments utilized. We have modified some of the included text on SECM results to hopefully help be more informative.

Minor Comments

L136: Why were these cutoffs chosen for the RNA seq experiment? Can the authors provide the corresponding P-values in Table S2 and S3?

The 2-fold cutoff is arbitrary but few genes are significantly differentially regulated below this value. Also, any phenotypic results for <2-fold changes in expression would be difficult to reproduce and outside of a few cases be of the lowest priority for further investigation. Our raw data and analysis methods are available for readers that would wish to investigate differentially expressed genes that fall below this cutoff. Table S3 has been removed based on reviewer comments below.

L161, 174, 196-7: The authors state "data not shown" in several instances.

For L174 we have modified this section with new data. L161 refers to qualitative data regarding a visible color change (red to yellow) which in our opinion did not justify the space required for another figure. L196-7 was not shown because the anaerobic RNASeq dataset for *C. matruchotii* is part of a separate ongoing investigation and catalase expression in that dataset would be a singular "0" value only. We opted to leave this out due to space and scope constraints that have already led us to split the work into two separate manuscripts. This data is also supported by the fact that the catalase mutant could not even be generated or grown aerobically, yet was made in our hands in anaerobic conditions and can grow anaerobically.

L242-63: Much of this text belongs in the methods section

We thank the reviewer for this suggestion. However, this section describes observational data recorded during SECM data collection for TEA+ and lactate ion profiles. Also, given the above major suggestion (#3) by the reviewer we are a bit conflicted on removing this given that it helps explain the SECM technique and could be contrary to the reviewer's wishes above. We have edited some of the text within the indicated areas and in the SECM section in general to help with clarity.

L317-9: If the transcriptional responses of *S. mitis* to *C. matruchotii* are part of a separate study, why are they included as Table S3?

We felt that these data would be of interest to the readers as it was acquired during the same RNASeq experiment as that from the *C. matruchotii* side. Given that we do not explore *S. mitis* responses to coculture conditions in this manuscript and to address this reviewer comment we have now removed these data.

Figure 2, 3: Include details in the legends indicating that these experiments were under aerobic conditions

Done, we thank the reviewer for this important clarification.

Figure 5: Can the authors report the data in panels B and C with respect to measured L-lactate concentration?

We appreciate the reviewers suggestion. We have updated the Fig. 5 legend to address this.

In the SECM image of Fig 5B, tip currents are dependent on not only TEA⁺ concentration in the bulk solution, but also a distance between a pipet tip and a glass substrate or a bacteria, and the permeability of bacterial membrane. The main reason for tip current changes over a bacteria is due to the hindered diffusion of TEA⁺ to the tip within a distance between a tip and nearly impermeable bacterial membrane, while TEA⁺ concentration exists almost uniformly outside of a bacteria. The conversion of tip current to TEA⁺ concentration is not feasible in this SECM image due to the non-linear relationship between the tip current and concentration of TEA⁺. Fig 5B is the SECM image probing only TEA⁺ not L-lactate.

In the SECM image of Fig 5C, the tip current is dependent on not only lactate concentration in bulk solution, but also a distance between a pipet tip and a glass substrate or a bacteria, the lactate concentration produced by *S. mitis*, and the rate of lactate consumption at *C. matruchotii*. Due to these multi-determining factors for tip currents and the non-linear relationship between the tip current and lactate concentration, the current scale in the SECM image cannot be straightforwardly converted to lactate concentration. This is why we do Finite Element Methods using COMSOL Multiphysics to solve a diffusion problem of lactate near a bacterium, and theoretically simulate the tip current-distance between a tip and a bacteria, and the lactate concentration profile over a single bacteria to accurately extract the information of lactate concentration above a single bacteria (Fig S2E and S2F) the lactate ranges are reported in the Fig 5 legend as ~0.5mM above the *S. mitis* surface and ~0.1mM above the *C. matruchotii* surface.

Figure S2: The legend indicates that this experiment was not replicated (i.e., N=1). The authors should replicate this experiment before reporting the results at L170.

This experiment has been updated with the now complemented lutABC mutant strain and replaced by table S1.

Line Comments

We thank the reviewer for these beneficial comments, unless directly indicated below, all changes were addressed at the relevant lines and changes can be observed in the track-changes manuscript copy provided for the main draft.

L36-7: What is meant by "directly adjacent in vivo"?

L44,157,272: Replace "1st" with "first"

L51: Be more clear about what "they" refers to in this sentence
L77,81: Avoid opening sentences with "It". Be more clear as to what "it" is referring to
L80: replace "with the" with "where the"
L80: replace "being" with "it"
L86-7: What is the "hedgehog" model? Why is it relevant for this work?

The citation for the model is provided upon its mention in the text. The model is the 1st description of the biogeography of *C. matruchotii* in supragingival plaque and details how it has numerous streptococci attached directly to it which was the initial inspiration for studying these polymicrobial interactions further.

L91: Spell out hydrogen peroxide fully during its first use
L93: Sentence has two instances of "which can"
L98, 136, 141, 171, 181, 391, 321, 332, 376, 393, 415: If a species name starts a sentence, then the genus should be written out fully
L100: add "-mediated" after *Corynebacterium*
L102,247-8: Enclose the i.e. statement in parentheses
L105-8: Recommend splitting into two sentences
L122: Would suggest including a preamble sentence before jumping into results
L130-1: Would move this sentence to the start of the following paragraph
L137-41: Would suggest splitting into two sentences
L184: Replace "data anaerobically" with "anaerobic data"
L195: Would suggest referencing Table S2 here
L203: What is "it" referring to?
L217: add "hypothesis" after "to test this..."
L218,219,221,222,224: The "WT" abbreviation has been defined
L310: add "led to" before "upregulation"
L361: What is "(KIM REF+)?"
L384-6: This sentence is repeated from L376-9
P411: Several instances of "will be" in this section

June 19, 2023

Dr. Matthew Mark Ramsey
University of Rhode Island College of the Environment and Life Sciences
Cell and Molecular Biology
120 Flagg Rd. CBLS Rm 387
Kingston, RI 02881

Re: mSystems00115-23R1 (Bacterial multispecies interaction mechanisms dictate biogeographical arrangement between the oral commensals *Corynebacterium matruchotii* and *Streptococcus mitis*)

Dear Dr. Matthew Mark Ramsey:

Thank you for submitting your manuscript to mSystems. We have completed our review and I am pleased to inform you that, in principle, we expect to accept it for publication in mSystems. However, acceptance will not be final until you have adequately addressed the editor comment below.

Preparing Revision Guidelines

Please return the manuscript within 60 days; if you cannot complete the modification within this time period, please contact me. If you do not wish to modify the manuscript and prefer to submit it to another journal, please notify me of your decision immediately so that the manuscript may be formally withdrawn from consideration by mSystems.

Sincerely,

Matthew Traxler

Editor, mSystems

Journals Department
Editor comments:

The work to complement the lutA- strain is appreciated. Please move table S1 to the main paper and revise accordingly.

To the editor and reviewers:

Please note that all responses to the reviewers are here in red and original review comments in black. We thank both reviewers for their requests and comments and we appreciate their time in reviewing and helping to improve this manuscript. We outline below our attempts to address all issues raised by writing and further experimentation. We would like to note that our companion manuscript regarding more of the operational details of the SECM technology used here has been accepted and now in press (DOI: 10.1021/acs.analchem.3c01498).

Also, all line numbers used in the response below refer to the original main draft “marked up” copy which is a track changes copy showing where text has been modified in response to reviewer comments. We again thank the reviewers and apologize if we were unable to completely satisfy every request given. We have done our best to explain any circumstances as to why we chose to present some information in the format here.

Reviewer #1 (Comments for the Author):

The manuscript describes a dual species interaction between two prominent and relevant members of the oral biofilm, *S. mitis* and *C. matruchotii*. The authors demonstrate that the metabolic exchange between both species is a benefitting factor in this dual species community. *S. mitis* is a producer of lactate and hydrogen peroxide, and *C. matruchotii* is able to metabolize lactate and detoxify hydrogen peroxide, thus counteracting the self inhibitory effect of hydrogen peroxide on *S. mitis*. Overall this manuscript is well written and highly relevant to oral biofilm development and provides a mechanistic explanation of species distribution. The usage of SECM to demonstrate metabolite exchange is highly innovative.

I only have **minor comments**:

For Fig. 5, it is not clear how representative this image is. Was this done multiple times? Is there some kind of quantification available?

Similar SECM images were reproducibly obtained over >3 coculture samples, where higher current over *S. mitis* and lower current over *C. matruchotii* than the background current indicate in situ production and consumption of lactate, respectively. In the main text (in page 7, line 254~275) and supporting information (in page S4 and Fig S3 in page S7), we mentioned that single *S. mitis* cells produce lactate up to 0.5 mM estimated by current in Fig 5C, theoretically simulated current-distance curves, and theoretically simulated concentration profile in Fig S3E and S3F. At the same time, *C. matruchotii* rapidly consumes lactate produced by *S. mitis* at a rate of $\geq 5 \times 10^6 \text{ s}^{-1}$, thereby decreasing the local concentration of lactate up to 0.1 mM over *C. matruchotii*. More detailed information and rigorous discussion about the theoretical simulation and the quantitative analysis should be found in the companion manuscript now accepted in Analytical Chemistry (10.1021/acs.analchem.3c01498) (submission copies provided to Editor). In the companion manuscript, we proved that only *S. mitis* produces lactate, while *C. matruchotii* doesn't produce any lactate in each monoculture study. Single cells of *S. mitis* in monoculture produce similar level of lactate to coculture sample in below Fig S2B and S2C (companion manuscript). In contrast, a single cell of *C. matruchotii* in monoculture does not produce any lactate, thus nothing appeared in the SECM image based on lactate ion transfer (Fig S3B,

companion manuscript). Herein, we share the additional SECM data (Fig S2 and S3) from the submitted companion manuscript as below for review purposes.

It would be nice to include a bigger picture in the discussion, including some of the other species that are known to metabolize lactic acid or detoxify hydrogen peroxide.

We thank the reviewer for this suggestion. Further discussion of peroxide production and lactate metabolism are included in lines 307-319.

Supplemental files:

page 4 line 147: should it read probe instead of problem?

We thank the reviewer for their careful attention!

It should be a two phase “problem” and we have corrected this error. Using COMSOL Multiphysics, we solved a diffusion problem of lactate between two phases of aqueous bulk phase and a single bacterium with laterally homogeneous membrane having a uniform permeability.

Fig. S2...growth curve should be in log-scale

We thank the reviewer for this correction, as part of other reviewer comments these data have now been replaced by table S1 in the supplement.

Reviewer #2 (Comments for the Author):

In this manuscript, Almeida and Puri et al. were interesting in characterizing metabolite-mediated interactions between the common oral commensal bacteria *Corynebacterium matruchotii* and *Streptococcus mitis*. To do so, the authors use a combination of traditional bacterial coculture and genetics, as well as a novel method for directly visualizing metabolic exchange between cells that they call scanning electrochemical microscopy (SECM). Together, Almeida and Puri determine that *C. matruchotii* is able to support to growth of *S. mitis* by detoxifying produced hydrogen peroxide and consuming produced lactate. The culture experiments are sound, but I have questions regarding the genetics and SECM methods.

Major Comments

1. The authors generate numerous mutants in both *C. matruchotii* and *S. mitis* throughout this manuscript, but they never complement the mutations in trans to confirm that their deletions are specifically responsible for the effects that they observe.

We have now generated and tested an *in trans* complement of *lutABC* to the original Δ *lutABC C. matruchotii* mutant. We verified here that the loss of growth rate in the *lutABC* mutant was complemented by addition of the *lutABC* operon *in trans* vs an empty vector Δ *lutABC C. matruchotii*. See lines 178-9, Materials and Methods and table S1. Also, please note the differences in growth rates from previous experiments were due to performing the new rate measurements in a different culture vessel type than previously.

The role of SpxB in H₂O₂ generation in *S. mitis* and other species has been previously well described which we discuss and provide further citations in response to reviewer 1 comments on lines 307-319. Likewise, complementation of *spxB* and restoration of H₂O₂ production in *S. mitis* has been previously described (<https://www.nature.com/articles/s41598-021-04562-4#Sec2>). We demonstrate here that removal of peroxide by exogenous catalase yielded a phenotypically similar result to that of the Δ *spxB S. mitis* (Fig. 3).

2. How are the authors sure that their SCEM method is specifically measuring L-lactate concentrations?

For instance, have the authors repeated their SCEM experiment using a *S. mitis* Δ *ldh* mutant? What about addition of an LDH inhibitor like oxamic acid? Alternatively, what happens if the authors image a coculture of *S. mitis* and a *C. matruchotii* mutant deleted *lutABC* and the two potentially reversible lactate dehydrogenase genes?

We independently measured the lactate ion transfer in aqueous bulk solution containing 2 mM lactate in Fig S3B, where lactate ion transfer is induced at ~ 0.42 V more positive than $E_{1/2}$ of TEA⁺ ion transfer. The limiting currents for lactate ion transfer respond to the lactate concentration linearly as well. This bulk electrochemistry confirmed the selective potential for lactate ion transfer across the pipet tip interface and confirms that we are detecting lactate.

Once we defined the potential for lactate ion transfer under nearly diffusion-limited condition in the bulk solution, we applied the same potential to the tip during raster-scanning above co-cultured bacteria. In the companion manuscript now in print at Analytical Chemistry (DOI 10.1021/acs.analchem.3c01498) (and provided to the editor), we demonstrated that only *S. mitis* produces lactate, while *C. matruchotii* does not. Single cells of *S. mitis* in monoculture produce similar level of lactate to coculture sample in Fig S2B and S2C (companion manuscript). In contrast, single cells of *C. matruchotii* in monoculture do not produce any lactate, thus nothing appeared in the SECM image based on lactate ion transfer (Fig S3B, companion manuscript). The relevant companion manuscript figures have been included above in replies to reviewer 1 above.

Regarding an *ldh* mutant given that we had already verified our ability to detect lactate specifically in sterile medium with and without lactate added, we did not see a rationale for inhibiting *ldh* after the fact. Further, given that Streptococcal metabolism is highly dependent on glycolysis and that they do not

respire, the likelihood of this mutant surviving much less behaving reproducibly in our SECM settings was unlikely thus it was not attempted. Lastly, we wish to note that we cannot differentiate between L or D lactate with SECM but that *C. matruchotii* does not appear to possess a lactate racemase nor D-lactate dehydrogenase.

3. Much of the Supplemental Results could be incorporated into the main manuscript text and would better inform readers as to how the SCEM method works

We thank the reviewer and agree with this comment. However, given the extensive nature of SECM work included it was already necessary to publish this in a second manuscript given the size constraints and technical nature of the SECM developments utilized. We have modified some of the included text on SECM results to hopefully help be more informative.

Minor Comments

L136: Why were these cutoffs chosen for the RNA seq experiment? Can the authors provide the corresponding P-values in Table S2 and S3?

The 2-fold cutoff is arbitrary but few genes are significantly differentially regulated below this value. Also, any phenotypic results for <2-fold changes in expression would be difficult to reproduce and outside of a few cases be of the lowest priority for further investigation. Our raw data and analysis methods are available for readers that would wish to investigate differentially expressed genes that fall below this cutoff. Table S3 has been removed based on reviewer comments below.

L161, 174, 196-7: The authors state "data not shown" in several instances.

For L174 we have modified this section with new data. L161 refers to qualitative data regarding a visible color change (red to yellow) which in our opinion did not justify the space required for another figure. L196-7 was not shown because the anaerobic RNASeq dataset for *C. matruchotii* is part of a separate ongoing investigation and catalase expression in that dataset would be a singular "0" value only. We opted to leave this out due to space and scope constraints that have already led us to split the work into two separate manuscripts. This data is also supported by the fact that the catalase mutant could not even be generated or grown aerobically, yet was made in our hands in anaerobic conditions and can grow anaerobically.

L242-63: Much of this text belongs in the methods section

We thank the reviewer for this suggestion. However, this section describes observational data recorded during SECM data collection for TEA+ and lactate ion profiles. Also, given the above major suggestion (#3) by the reviewer we are a bit conflicted on removing this given that it helps explain the SECM technique and could be contrary to the reviewer's wishes above. We have edited some of the text within the indicated areas and in the SECM section in general to help with clarity.

L317-9: If the transcriptional responses of *S. mitis* to *C. matruchotii* are part of a separate study, why are they included as Table S3?

We felt that these data would be of interest to the readers as it was acquired during the same RNASeq experiment as that from the *C. matruchotii* side. Given that we do not explore *S. mitis* responses to coculture conditions in this manuscript and to address this reviewer comment we have now removed these data.

Figure 2, 3: Include details in the legends indicating that these experiments were under aerobic conditions

Done, we thank the reviewer for this important clarification.

Figure 5: Can the authors report the data in panels B and C with respect to measured L-lactate concentration?

We appreciate the reviewers suggestion. We have updated the Fig. 5 legend to address this.

In the SECM image of Fig 5B, tip currents are dependent on not only TEA⁺ concentration in the bulk solution, but also a distance between a pipet tip and a glass substrate or a bacteria, and the permeability of bacterial membrane. The main reason for tip current changes over a bacteria is due to the hindered diffusion of TEA⁺ to the tip within a distance between a tip and nearly impermeable bacterial membrane, while TEA⁺ concentration exists almost uniformly outside of a bacteria. The conversion of tip current to TEA⁺ concentration is not feasible in this SECM image due to the non-linear relationship between the tip current and concentration of TEA⁺. Fig 5B is the SECM image probing only TEA⁺ not L-lactate.

In the SECM image of Fig 5C, the tip current is dependent on not only lactate concentration in bulk solution, but also a distance between a pipet tip and a glass substrate or a bacteria, the lactate concentration produced by *S. mitis*, and the rate of lactate consumption at *C. matruchotii*. Due to these multi-determining factors for tip currents and the non-linear relationship between the tip current and lactate concentration, the current scale in the SECM image cannot be straightforwardly converted to lactate concentration. This is why we do Finite Element Methods using COMSOL Multiphysics to solve a diffusion problem of lactate near a bacterium, and theoretically simulate the tip current-distance between a tip and a bacteria, and the lactate concentration profile over a single bacteria to accurately extract the information of lactate concentration above a single bacteria (Fig S2E and S2F) the lactate ranges are reported in the Fig 5 legend as ~0.5mM above the *S. mitis* surface and ~0.1mM above the *C. matruchotii* surface.

Figure S2: The legend indicates that this experiment was not replicated (i.e., N=1). The authors should replicate this experiment before reporting the results at L170.

This experiment has been updated with the now complemented lutABC mutant strain and replaced by table S1.

Line Comments

We thank the reviewer for these beneficial comments, unless directly indicated below, all changes were addressed at the relevant lines and changes can be observed in the track-changes manuscript copy provided for the main draft.

L36-7: What is meant by "directly adjacent in vivo"?

L44,157,272: Replace "1st" with "first"

L51: Be more clear about what "they" refers to in this sentence

L77,81: Avoid opening sentences with "It". Be more clear as to what "it" is referring to

L80: replace "with the" with "where the"

L80: replace "being" with "it"

L86-7: What is the "hedgehog" model? Why is it relevant for this work?

The citation for the model is provided upon its mention in the text. The model is the 1st description of the biogeography of *C. matruchotii* in supragingival plaque and details how it has numerous streptococci attached directly to it which was the initial inspiration for studying these polymicrobial interactions further.

L91: Spell out hydrogen peroxide fully during its first use

L93: Sentence has two instances of "which can"

L98, 136, 141, 171, 181, 391, 321, 332, 376, 393, 415: If a species name starts a sentence, then the genus should be written out fully

L100: add "-mediated" after *Corynebacterium*

L102,247-8: Enclose the i.e. statement in parentheses

L105-8: Recommend splitting into two sentences

L122: Would suggest including a preamble sentence before jumping into results

L130-1: Would move this sentence to the start of the following paragraph

L137-41: Would suggest splitting into two sentences

L184: Replace "data anaerobically" with "anaerobic data"

L195: Would suggest referencing Table S2 here

L203: What is "it" referring to?

L217: add "hypothesis" after "to test this..."

L218,219,221,222,224: The "WT" abbreviation has been defined

L310: add "led to" before "upregulation"

L361: What is "(KIM REF+)?"

L384-6: This sentence is repeated from L376-9

P411: Several instances of "will be" in this section

June 28, 2023

Dr. Matthew Mark Ramsey
University of Rhode Island College of the Environment and Life Sciences
Cell and Molecular Biology
120 Flagg Rd. CBLR Rm 387
Kingston, RI 02881

Re: mSystems00115-23R2 (Bacterial multispecies interaction mechanisms dictate biogeographical arrangement between the oral commensals *Corynebacterium matruchotii* and *Streptococcus mitis*)

Dear Dr. Matthew Mark Ramsey:

Your manuscript has been accepted, and I am forwarding it to the ASM Journals Department for publication. For your reference, ASM Journals' address is given below. Before it can be scheduled for publication, your manuscript will be checked by the mSystems production staff to make sure that all elements meet the technical requirements for publication. They will contact you if anything needs to be revised before copyediting and production can begin. Otherwise, you will be notified when your proofs are ready to be viewed.

If you would like to submit a potential Featured Image, please email a file and a short legend to msystems@asmusa.org. Please note that we can only consider images that (i) the authors created or own and (ii) have not been previously published. By submitting, you agree that the image can be used under the same terms as the published article. File requirements: square dimensions (4" x 4"), 300 dpi resolution, RGB colorspace, TIF file format.

We recognize that the video files can become quite large, and so to avoid quality loss ASM suggests sending the video file via <https://www.wetransfer.com/>. When you have a final version of the video and the still ready to share, please send it to mSystems staff at msystems@asmusa.org.

Sincerely,

Matthew Traxler
Editor, mSystems

Journals Department
E-mail: mSystems@asmusa.org